# Detecting face presentation attacks in mobile devices with a patch-based CNN and a sensor-aware loss function

**Waldir R. Almeida**[1], **Fernanda A. Andaló**[1]*, **Rafael Padilha**[1], **Gabriel Bertocco**[1], **William Dias**[1], **Ricardo da S. Torres**[2], **Jacques Wainer**[1], **Anderson Rocha**[1]

**1** Institute of Computing, University of Campinas, Campinas, São Paulo, Brazil, **2** Department of ICT and Natural Sciences, Faculty of Information Technology and Electrical Engineering, NTNU, Ålesund, Norway

* feandalo@ic.unicamp.br

## Abstract

With the widespread use of biometric authentication comes the exploitation of presentation attacks, possibly undermining the effectiveness of these technologies in real-world setups. One example takes place when an impostor, aiming at unlocking someone else's smartphone, deceives the built-in face recognition system by presenting a printed image of the user. In this work, we study the problem of automatically detecting presentation attacks against face authentication methods, considering the use-case of fast device unlocking and hardware constraints of mobile devices. To enrich the understanding of how a purely software-based method can be used to tackle the problem, we present a solely data-driven approach trained with multi-resolution patches and a multi-objective loss function crafted specifically to the problem. We provide a careful analysis that considers several user-disjoint and cross-factor protocols, highlighting some of the problems with current datasets and approaches. Such analysis, besides demonstrating the competitive results yielded by the proposed method, provides a better conceptual understanding of the problem. To further enhance efficacy and discriminability, we propose a method that leverages the available gallery of user data in the device and adapts the method decision-making process to the user's and the device's own characteristics. Finally, we introduce a new presentation-attack dataset tailored to the mobile-device setup, with real-world variations in lighting, including outdoors and low-light sessions, in contrast to existing public datasets.

## Introduction

Smartphones have become so popular that they are almost an extension of the user's body and mind. Most people use them as their main medium of communication, storing conversational history, pictures, passwords, and other private data. As such, these devices must be secured, so that only the owner can access the data stored therein. Face authentication is a convenient way for unlocking such devices, requiring only that the owner looks at the built-in frontal camera, as in normal usage.

**Data Availability Statement:** We have uploaded our dataset to the Zenodo repository (https://zenodo.org/record/3749309#.Xv5Hr2gzZ3h). The dataset is now published and is available under an

EULA, required by the Ethics Committee that approved the collection of the data.

**Funding:** This work was funded by Motorola. The funders had no role in study design, data collection and analysis, decision to publish, or preparation of the manuscript.

**Competing interests:** The authors have declared that no competing interests exist.

However, it has become popular knowledge that face authentication systems are somewhat vulnerable to presentation attacks (PA) at the sensor level. A PA can be made by simply showing the system an image of the device owner. That requires little technical expertise, as most people's images are readily available on the Internet, and even a laptop display could be used as an attack medium.

Over the last years, there have been increased research interest in face presentation attack detection (PAD), but the existing approaches have been shown not to generalize beyond the conditions represented in the public datasets used as benchmarks, as is evident in cross-dataset evaluations [1–4]. Moreover, most studies are based on similar handcrafted features, and do not target the mobile-device scenario.

In this work, we focus on face PAD for modern smartphones, considering printed-photo and screen attacks. We take a data-driven approach and present training techniques targeting the PAD problem. We adapt a pre-trained architecture for PAD using, during training, multi-resolution face patches, making the model more robust to changes in resolution, while also avoiding overfitting to specific facial features. We introduce a loss function that closely models the PAD objective, forcing genuine-access examples from the same device to be more compactly located in the learned feature space, while also reducing inter-device confusion. By using a lightweight but powerful architecture as the core of the proposed method, we ensure that inference can run with small memory footprint and in under one second in modern smartphones.

To further improve the effectiveness of these models in real-world situations, when they are deployed on mobile devices and are presented with images from the same user, we also propose one strategy to adapt the decision boundary to the characteristics of a specific user and of a sensor device.

Our contributions are the following:

- Two techniques to train deep convolutional neural networks to model the problem in a purely data-driven fashion, with RGB pixels as input.

- A simple yet effective method for adapting a trained model by using a gallery of user data on the device thus heightening the discriminability of the model.

- A novel face presentation-attack detection dataset—RECOD-MPAD—that is representative of the target scenario herein, with more realistic illumination conditions.

- An extensive study of error cases, considering multiple factor-disjoint protocols, which leads to a better understanding of the problem.

The remainder of this article is organized as follows. The background section explores important concepts and outlines several PAD methods in the literature. The proposed method section presents our approach to tackle the PAD problem, with a detailed description of the proposed method and its techniques. The datasets section describes the datasets used in the experiments, highlighting the one specifically constructed for this work. The experimental results section analyzes and validates the methods in terms of performance and comparative experiments, considering multiple factor-disjoint protocols. Finally, the last section draws conclusions and presents possible future directions of investigation.

## Background

We start by looking at how genuine-access and attack images are created, discussing some general assumptions that are involved in software-based presentation-attack detection. Next, we

present an overview of some relevant techniques in the literature. Finally, we expose some of the problems with the state of the art, motivating our approach.

## Image acquisition and attack clues

The presentation attack detection problem consists of answering whether or not a captured biometric sample is genuine. Note that the only resource is the biometric sample, and the hypothesis is that we can answer the question by looking at pixels only. To seek for attack clues and understand how the problem may be solvable, we take into consideration how image data is transformed before being acquired by the camera in the user device.

While genuine samples are acquired as a single capture by directly photographing the authenticating user, in an attack event the biometric sensor actually recaptures a previously captured image of the user, which is displayed on an attack instrument (paper or screen); i.e., an "attack camera" captures and preprocesses an image of the target user face and, when that image is displayed on the attack instrument, it is further modulated by the medium's own reproduction, geometric, and reflectance characteristics.

Each part of the recapturing process changes the data to different degrees, and it is not trivial to identify whether an image feature is due to the interfering attack camera or display medium, indicating an attack, or simply a normal variation in the user facial traits or lighting conditions during acquisition. All factors can vary arbitrarily and interact in seemingly unforeseeable ways.

In comparison to a similar genuine image, attacks often have different color distributions, due to limitations of the reproduction medium. Overall contrast is typically lower in printouts, due to soft focus and the influence of the light source on the flat surface, and higher in most electronic displays, due to the strong backlight. Printouts can have visible printing defects, while low-quality liquid-crystal displays (LCD) can suffer from varying brightness levels throughout the screen. Finally, the resampling process often generates its own aliasing artifacts. One example is the moiré pattern [5] that appears when a sensor samples images containing fine-grained regular structures, such as the pixel grid in electronic displays. Other regular artifacts can be caused by slow refresh rates in older displays or low frame rates when replaying videos.

## An overview of existing methods

Over the last years, different PAD methods were proposed. The interest increased significantly since the release of the NUAA dataset [6] and the advent of the first competitions. Because of that, it is no longer feasible to give an exhaustive analysis of all published methods. It is, however, noticeable that most methods are related, and tend to be based on common assumptions and feature descriptors.

**Based on liveness or motion detection.** These methods seek to detect PAs through evidence for the lack of vitality in the captured face and typically depend on motion information. The archetypal method is eye-blink detection [7], which can be effective if the attacker uses a photograph, but is easily circumvented by video replay attacks, or even by cutting holes in the printed face image and using one's own eyes to simulate blinking [8]. Another class of methods tries to detect subtle movements of a living human face, using optical-flow estimation [9], motion magnification [10], or temporal extensions of low-level texture descriptors [11]. Some methods take advantage of motion correlations between foreground and background or other scenic clues [12]. This is likely to succeed if the attacker does not use a fixed-support when performing the attack with a printout or display, but would probably fail otherwise. These methods have the disadvantage of requiring a potentially long sequence of frames to make a single

prediction, and most of them can be circumvented by faking eye-blinks and carefully handling the attack instruments.

**Based on physics or geometry.**   Face PAs typically present the forged user representation on a flat surface, which has different reflectance properties compared to a living face. Some methods seek to detect this "flatness" or abnormal reflectance with physical or geometric motivations. One early method tries to capture depth information via Structure-from-Motion techniques [13]. Others propose to detect differences in motion between face areas via optical flow estimation [14] or by explicitly modeling 3D projective invariants [15]. Another possibility is to model local curvatures by using multiple images [16]. These methods typically require at least some user cooperation to succeed.

By assuming a simplified Lambertian model of reflectance, it is also possible to model the interaction between the illuminant and the reflective surface to extract albedo and normal maps [6], which are then used as representations to discriminate genuine-access from attack samples. Although the motivation is clear, lighting in the real-world is mixed and uncontrolled, so the basic assumptions do not hold in practice. Another option is to model the diffuse and specular components to try to separate the latter, which could emphasize characteristics of the attack medium surface [17].

**Based on texture, noise analysis, or image quality.**   These methods seek to detect artifacts left by the recapture process or estimate degradation in overall image quality. Texture characterization is typically motivated as a means of discriminating the intrinsic textural properties of attack instruments and living faces, but can also capture other types of high-frequency information. Most are based on variations of local-binary pattern (LBP) descriptors [3, 18], but temporal extensions were also proposed [11]. Other methods use a combination of low-level local descriptors [19]. Frequency-specific information can be captured by Difference-of-Gaussians (DoG) filtering [8] or through Fourier analysis [20, 21].

A more global characterization that discards content information in static-content videos to analyze noise signatures is proposed in [22], while the same type of residual information is encoded as mid-level temporal representations in [2]. Methods based on low-level texture descriptors or high frequency information can be effective in detecting paper texture and noise patterns, however the effectiveness is extremely dependent on the exact acquisition conditions and the capability of the camera resolving fine details. Moiré-like patterns are strong clues for attacks, but are not always present, making countermeasures solely based on them unreliable.

Explicit attempts at capturing image distortion artifacts can be found in [4]. Researchers also explored generic image quality metrics directly [23, 24]. As some of these metrics require a reference image, which is not available, these works compare the probe image to an artificially degraded version of itself. The hypothesis is that the difference is greater between genuine-access images than between attack images, since the latter are assumed to be of lower-quality, which is not always valid. Although under similar acquisition conditions attack and genuine-access samples would potentially be separable by generic image quality metrics and statistics, existing algorithms do not take context into consideration, which makes them fragile in real-world scenarios.

**Based on feature learning.**   Since 2012 [25], models based on Convolutional Neural Networks (CNNs) have achieved state-of-the-art performance on many image recognition tasks. Data-driven methods like these, which receive pixels as input and learn intermediate representations directly from data, are said to perform *representation* or *feature learning* [26]. Feature learning is underrepresented in the face PAD literature, despite its success in other visual tasks.

Menotti et al. [27] studied architecture optimization for PAD. In this strategy, many simple architectures with random convolutional filters are sampled and used as feature extractors to

train a final linear classifier. They found out that, although competitive, optimized architectures could not be improved by having their parameters further adjusted. This could be partially attributed to insufficient hyperparameter tuning involved in Stochastic Gradient Descent (SGD) training.

Yang et al. [28] trained a CNN based on the *AlexNet* network [25], using a classifier at the end. During pre-processing, they experimented with a few face-centered regions, including tighter face crops, and regions showing more background. They reported promising results on different datasets, but the best pre-processing configuration was different in each case. As one of the conclusions, they highlight differences in background between the two datasets, which makes evident that the network learned to exploit acquisition biases when too much background was used in training.

Patel et al. [29] also experimented with training deep CNNs using aligned faces and the whole frame as input. This choice is not well-founded, since it is strongly dependent on the dataset. The final system consists of a fusion scheme involving the output of the CNN and an eye-blink detector.

Atoum et al. [30] introduced a two-stream CNN for PAD, by extracting local features from patches and constructing depth maps from face images. The use of patches makes the method independent of spatial face areas and depth maps can be used to detect the presence of face-like depth. This work, differently from ours, does not consider multi-resolution patches to increase robustness neither a custom-tailered loss function to the PAD problem.

Jourabloo et al. [31] revisit the analysis of residual information for PAD by training a CNN to estimate recapture-related noise in order to create live faces. Other works formulate the PAD problem differently, aiming at creating more discriminative and generalizable representations. Liu et al. [32] argue about the importance of auxiliary supervision, instead of considering the PAD problem simply as a binary classification. Li et al. [33] take both spatial and temporal information by considering a 3D CNN which is firstly trained with cross-entropy loss, and further enhanced with a generalization loss.

We highlight that our approach falls into this category, however, we specifically target the mobile-device scenario. Critically, none of the relevant related work considers modern datasets for that scenario, and so far strategies for using the available data and training the networks have been limited.

## A critical look at the state of the art

Early methods for face PAD were mostly based on eye-blink detection and other motion clues, which require several frames to be acquired, and typically fail under video replay attacks or simple cut-photo attacks. The community then moved on to exploring potentially more generalizable clues based on texture description, but currently most of these methods are based on the same low-level descriptors and simple classifiers, and yet they were shown to fail under more challenging cross-dataset protocols [1]. Other recent methods also suffer from the same problem [2–4].

Inasmuch as available public datasets have been useful for comparing different approaches, and inspiring new research efforts, they are now mostly outdated, both in terms of available cameras and attacks, and in terms of methodology. The partial shift to cross-dataset evaluations has shown the limitations of methods and datasets alike. Only recently has the community started to address the specific constraints of mobile applications [4]. New datasets, such as OULU-NPU [34] and REPLAY-MOBILE [35], have appeared with accompanying modern protocols, but they still have some of the same problems as other datasets, such as static sessions with low illumination variability.

Finally, efforts in devising deep learning, or other data-driven approaches to face PAD have been limited, with most solutions based on very similar aligned-face pre-processing and training strategies, and not taking into account our constraints (image acquisition peculiarities and limited memory and processing power). To the best of our knowledge, there are no rigorous studies of such methods considering modern protocols and mobile devices. It is in this context that we propose to study the problem in a purely data-driven fashion, aiming at gaining insight into how far we can go with software-based methods in such scenario.

## Proposed method

We propose a method based on training a Convolutional Neural Network (CNN) to distinguish between genuine-access and attack images. Comparing to the traditional way of training a CNN with whole-face images and a cross-entropy loss, the proposed techniques change what a CNN sees as input during training, and how it is optimized at the end. This makes sense from a modeling perspective, because the interaction between the input and the optimization objective is what really defines the problem, driving the learning procedure.

We take inspiration from one of the versions of the SqueezeNet architecture [36]. As a baseline, we adapt the architecture to consider presentation-attack detection, formulating the problem as a 2-class classification, and using only aligned whole-face images for training the network. From this modified architecture, we introduce our method: the use of face patches of variable resolution during training, which reduces overfitting to user-specific characteristics, and promotes the learning of more robust representations that are not tied to a single scale; and a loss function that more closely models the PAD objective, promoting the compactness of intra-device genuine examples in the learned feature space.

### Convolutional neural network core architecture

We adapt SqueezeNet v1.1 architecture [36] as our core architecture for different reasons: firstly, the network is small and fast enough to be embedded in mobile devices, in contrast to other popular architectures [37, 38]; it has a fully-convolutional structure, making interpretation of results easier, and is flexible to changing input size and alignment; finally, it was proven to be more accurate than AlexNet, which validates its potential for representing complex visual relationships.

The network is illustrated in Fig 1. Next to each arrow, we show the shape of the output tensor for a single input image of size $3 \times 227 \times 227$. For instance, the fire module *fire4* receives as input 128 activation maps of spatial dimension $28 \times 28$ and outputs 256 maps of size $28 \times 28$. Fire modules (illustrated at the top-right corner) are similar to inception modules [39] and are the main building blocks of SqueezeNet. They massively reduce the total number of parameters in $3 \times 3$ convolutions by first squeezing the channel dimension of the input tensor with $1 \times 1$ convolutions. The classification layer consists of a dropout operation [40] to reduce overfitting and a convolutional layer producing a number of feature maps that matches the number of classes in ImageNet. By averaging each class-specific map individually, the network also reduces the total number of parameters by eliminating the need for fully-connected layers. In summary, SqueezeNet v1.1 has approximately 1.2 million parameters, which can be stored in less than 5MB of memory, making it suitable to be used in mobile devices.

### Baseline: CNN training with whole-face images

The baseline we adopt herein considers training a deep CNN by using aligned whole-face images, the traditional input format in most algorithms published in the literature [27]. In this case, however, the pipeline consists mostly of the whole multi-layered network, which is

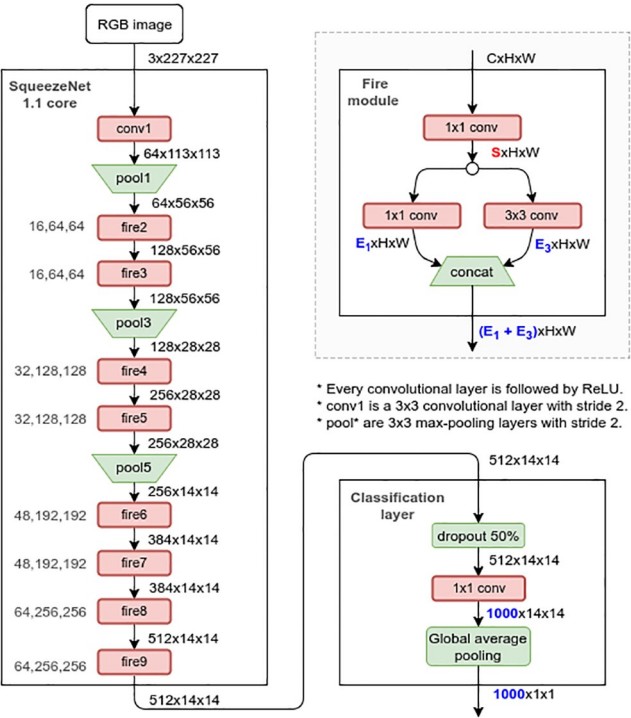

**Fig 1. Squeezenet v1.1.** Original architecture [36] and generic micro-architectural details of a fire module.

trained from end to end. Using aligned whole-face images can be justified as a means of reducing unnecessary variations during training and inference, putting the data in a predictable content-domain.

**Architecture.** The core architectural component is SqueezeNet (Fig 1). As it is a fully-convolutional network, all feature maps are flexible in size, but we keep the input size as $227 \times 227$. Since the cropped-face region in our scenario typically varies from 300 to 550 pixels in each dimension, keeping the original input size ensures that only small details are lost due to rescaling, while increasing it would make the network too slow to train and be used in mobile devices. Most PAD pipelines in the literature pre-process images to a fixed size, typically varying from $64 \times 64$ to $256 \times 256$.

**Pre-processing and data augmentation.** We start from an aligned and square-cropped image of the face region, both in training and inference phases. In practice, we found that the exact alignment does not significantly impact the performance of the method. During training, we read an RGB image, rescale the aligned face region to $256 \times 256$, crop a $227 \times 227$ central region, and flip the image horizontally with probability 0.5. Other data augmentation strategies that involve random photometric distortions and normalization [25] are potentially destructive for label information.

Before feeding the image into the network, we perform a simple pixel-wise transformation from the range [0.0, 1.0] to the range [− 1.0, 1.0]. Basic centering is common-place to obtain meaningful gradients in the first iterations of training, especially if parameters are randomly initialized and the ReLU activation function is used [25, 37]. In practice, we found that this transformation does not significantly affect accuracy, but it helps to make the training procedure more stable.

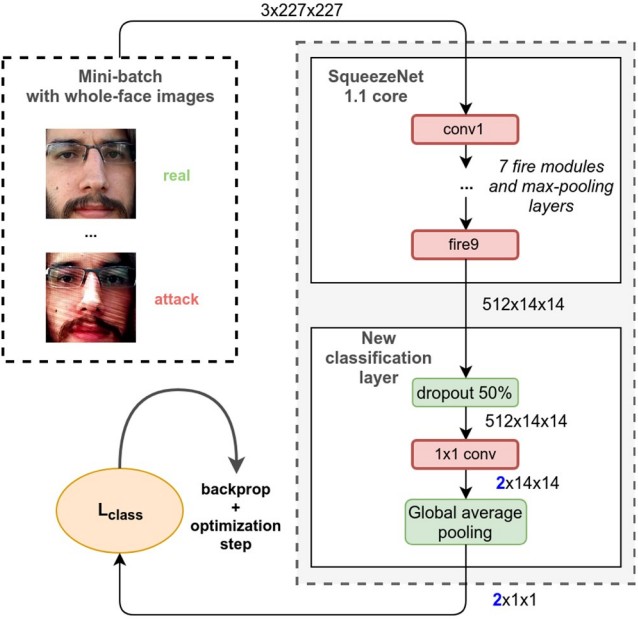

**Fig 2. Baseline (*Whole-face CNN*).** Architecture and training procedure.

**Training details.**   Training is done via standard backpropagation [41]. Fig 2 illustrates the architecture and the training procedure. We feed the network with a preprocessed mini-batch of images containing faces and their labels. In each iteration, 64 images are randomly sampled with replacement from the training set. The probability of a single image being selected is inversely proportional to the number of samples with its label in the training set, to account for class imbalance. Each mini-batch consists of roughly 32 samples with label *genuine* and 32 samples with label *attack*.

The 2-dimensional output corresponding to the two classes is used as input to a cross-entropy criterion, which is analogous to a traditional Softmax classifier. The function can be interpreted as normalizing the input vector into probabilities, and then measuring the mismatch between the predicted distribution and the expected distribution in which the mass is fully concentrated in the true label. In practice, we average over the whole mini-batch, giving the following expression, where $f_c(X)$ is the network output for class $c$ and input $X$, and $B$ is a mini-batch of training examples:

$$L_{class}(B) = \frac{1}{|B|} \sum_{(X,y) \in B} - \log \left( \frac{e^{f_y(X)}}{e^{f_0(X)} + e^{f_1(X)}} \right). \qquad (1)$$

After computing the loss and intermediate activations, the gradient of the loss with respect to every adjustable parameter is computed via backpropagation. Finally, for the optimization step, we use the Adam optimizer [42], which is an adaptive optimizer based on SGD with momentum, requiring minimal hyperparameter tuning. All experiments were carried out with default Adam hyperparameters and a learning rate of $10^{-5}$. As regularization, we add to the loss function an L2 penalty (weight decay) with weight $10^{-4}$.

For parameter initialization, we start from pre-trained ImageNet weights for the core part of the network, which is preferable to random initialization. For the classification layer, we initialize biases to 0.0, and weights from a normal distribution with mean 0.0 and standard deviation 0.01.

**Inference.** After the network is trained, it can be used to infer the label of new input images. Pre-processing is mostly as in the training phase. The detected face region is rescaled to $256 \times 256$ and centrally cropped to $227 \times 227$. The aligned and cropped whole-face image is centered in the pixel space by subtracting 0.5 from every pixel, and dividing by 0.5. In contrast to the training phase, no random mirroring is performed.

## CNN training with multi-resolution patches and a multi-objective loss function

Our proposed method first models the problem as a task of distinguishing regions of arbitrary level of detail in attack images from regions of arbitrary level of detail in genuine images. We accomplish this in training by extracting patches of varying sizes from the full-resolution images, only then rescaling them to the network input format.

This approach is beneficial in different ways. Firstly, it increases the number of examples available for training, taking full advantage of the training data by not discarding information that would be lost by premature re-scaling. By forcing the network to distinguish patches at different resolutions, its robustness to blur, adverse lighting, and unseen cameras is increased. Finally, by not always receiving the whole user face, the network is encouraged not to depend on user-specific characteristics, which potentially reduces over-fitting.

The method then contemplates the problem of training models to be sensitive to a wide number of attack clues and sensor device specificities. We may ponder what is the best way to account for these differences during training or if it is reasonable to assume that genuine samples from different devices should have similar characteristics.

By analyzing overall noise and persistent high-frequency information across classes and sensors, we can observe more subtle differences between genuine and attack samples than between samples acquired by different devices. In Fig 3, we can observe that pattern noise can be more similar between genuine and attack samples than genuine access samples from different devices. This indicates that cameras are very distinct from each other and a formulation that does not account for their difference may end up with a model biased towards irrelevant aspects of the dataset, instead of representing important characteristics of the problem, such as attack clues. Inspired by this observation, we propose a loss function aiming at reducing such possible biases.

We reformulate the problem by adding another term to the training objective loss function, changing the way images are used during optimization. The goal is to force genuine samples *from a given device* to be more compactly located in intermediate feature spaces, but farther away from attack samples of the same device. We hypothesize that this would create better manifolds by not directly confounding information from different devices, as in traditional training strategies.

More specifically, we consider a latent representation $f_\ell(I)$ of the original input image $I$ after it has been successively non-linearly transformed by $\ell$ layers. Consider a triplet of images $I_n, I_r, I_a$ *coming from the same device*: a genuine *anchor*, another genuine example, and an attack example, respectively. Let $n := f_\ell(I_n)$, $r := f_\ell(I_r)$, and $a := f_\ell(I_a)$, for short. Now, we can add the following loss function to the network:

$$L_{triplet}(\mathbf{n}, \mathbf{r}, \mathbf{a}) = \max\left(||\mathbf{n} - \mathbf{r}||_2^2 - ||\mathbf{n} - \mathbf{a}||_2^2 + m, 0\right), \qquad (2)$$

where $m$ is a margin hyperparameter, fixed beforehand, interpreted as the relative separation between attack examples and genuine examples to be enforced in the learned embedding. In general, this separation should be as large as possible, but when $m >> 0$, training becomes one-sided, since the objective is reduced to making attack examples as separated from its

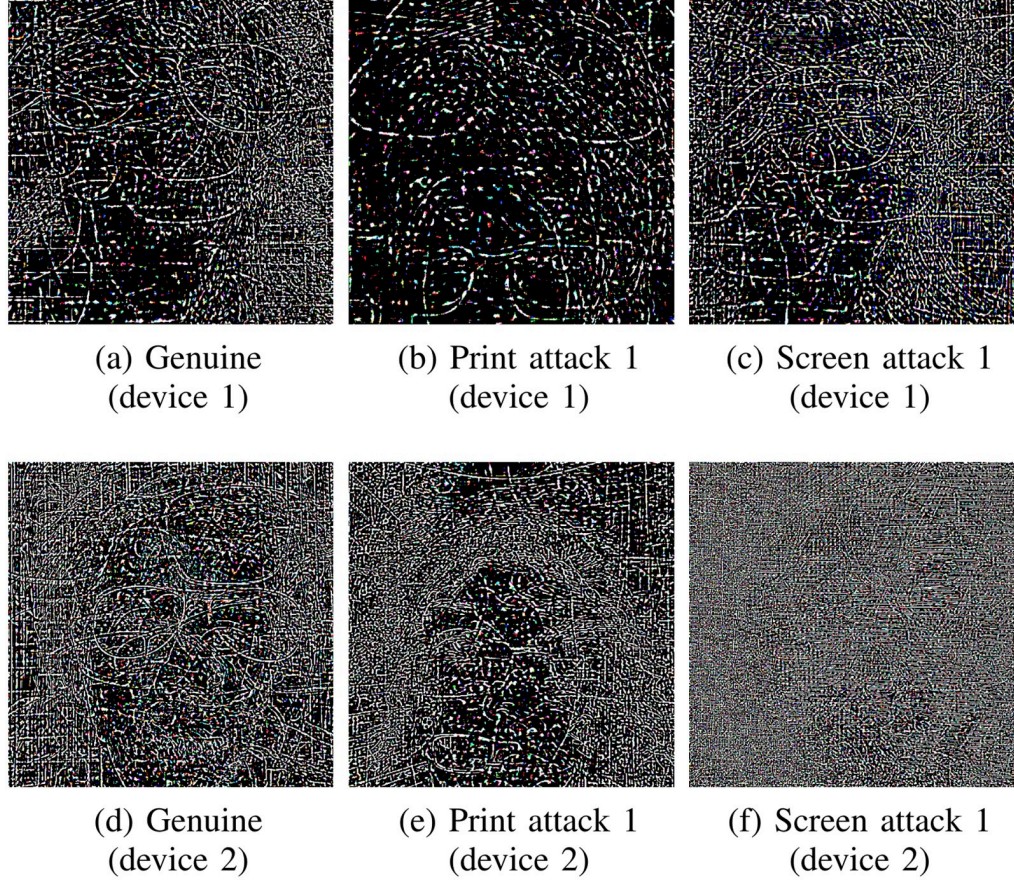

(a) Genuine
(device 1)

(b) Print attack 1
(device 1)

(c) Screen attack 1
(device 1)

(d) Genuine
(device 2)

(e) Print attack 1
(device 2)

(f) Screen attack 1
(device 2)

**Fig 3. Center-cropped noise residuals for average frames from RECOD-MPAD dataset, highlighting differences in pattern noise across sensors.** In each case, 20 frames were randomly sampled from the training set and the residual [43] was computed from the average frame. Despite belonging to different PAD classes, patterns are visually similar between (a) and (c), and between (d) and (e). On the other hand, genuine-access examples can generate patterns that look dissimilar when comparing across sensors, as in (a) compared to (d). As sensor devices are different and interact differently with attack instruments, this difference should be taken into account to train more robust data-driven models for PAD.

anchor as possible. Typically, training diverges due to large initial gradients, unless the learning rate is also reduced. On the other hand, the absence of such hyperparameter ($m = 0$) can prevent the network from learning the "attack concept".

By minimizing $L_{triplet}$, we enforce the notion that genuine samples from a given device should be closer in this latent space to genuine samples of the same device sensor, but farther away from attack samples of the same device, up to a margin. In theory, this triplet loss could be used alone to optimize an embedding to directly compare pairs of images during inference [44]. But since we ultimately want the trained model to distinguish between arbitrary genuine-access images and attack images, we add the previously described cross-entropy loss to jointly enforce a classification objective:

$$L_{spoof}(N, R, A) = \kappa \max_{1 \leq i \leq T} [L_{triplet}(\mathbf{n_i}, \mathbf{r_i}, \mathbf{a_i})] + L_{class}(f(R \cup A)),  \quad (3)$$

where $N$, $R$, and $A$ are the sets of genuine anchors, genuine non-anchor images, and attack images in a mini-batch, respectively, and $T$ is the number of triplets. All images in a mini-batch are sampled from a single device. $L_{class}(f(R \cup A))$ indicates the cross entropy loss is computed only for the $2T$ non-anchor images. In practice, the embedding term is computed as the

maximum triplet loss over triplets in a mini-batch. This can be interpreted as performing an online selection of hard triplets [44]. The hyperparameter $\kappa$ controls the relative weight of the triplet loss term.

Each term is a regularizer for the other objective in the spoof loss function $L_{spoof}$. The classification loss $L_{class}$ encourages finding a single decision boundary separating genuine-access from attack patterns in general, while the triplet loss $L_{triplet}$ encourages making intra-device genuine-access samples as compactly located as possible in the latent space, but farther away from attack samples of the same device.

**Architecture.** The core architecture is the same as the baseline, but we anticipate global-average pooling to directly reduce spatial correlations and the dimension of the activations of the *fire9* layer. This is the representation we use to compute the triplet loss. Following the 512-D embedding, we include the usual classification layer consisting of dropout and a linear mapping that generates class scores by optimizing the classification loss.

The network core consists mostly of stacked convolutional filters, which can be somewhat independent of scale. Different combinations of these filters, acting as more advanced feature detectors, can naturally specialize or learn to adapt to the varying resolutions and feature scales. Moreover, the final classification layer implemented as global-average pooling acts as a parameterless aggregator of the final filter responses, and is thus invariant to locality in its input representation.

Fig 4 illustrates how the network is trained with the multi-objective loss $L_{spoof}$. Mini-batches are built from triplets of images coming from the same device. Three different columns are formed, one with genuine-access anchors, one with genuine-access samples, and one with attack samples. The intra-device triplet loss $L_{triplet}$ is calculated for each triplet. In addition to

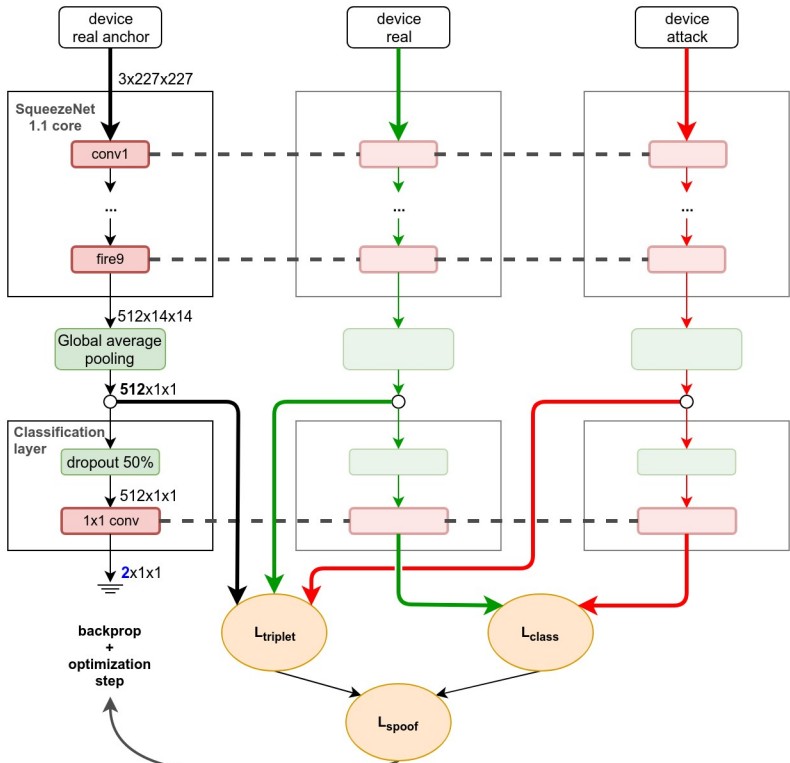

**Fig 4. Proposed method.** Architectural changes and training procedure.

that, the non-anchor columns are forwarded further into the classification layer, and their output is used to compute the usual cross-entropy loss $L_{class}$. Our *spoof loss* $L_{spoof}$ is the weighted sum of these two loss components, with the triplet-loss component weighted by a parameter $\kappa$. The dashed lines indicate that weights are shared between the columns, i.e., we train a single network.

**Pre-processing and data augmentation.** Fig 5 illustrates the construction of a mini-batch for multi-resolution patches. Pre-processing for each image in a mini-batch starts by uniformly sampling a variable $\alpha$ from the interval [0.08, 1], defining a percentage of the image area. The size of the cropped region is then $S = \sqrt{\alpha WH}$, where $W$ and $H$ are the width and height of the full-resolution whole-face image, respectively. The smallest possible patch corresponds to an area approximately equal to the region around one of the eyes in the original aligned image, regardless of the image size. A patch of size $S \times S$ is then cropped from randomly sampled top-left corners $i, j$. The remaining pre-processing and augmentation procedure is similar to the baseline: rescaling of the patch to $227 \times 227$, random mirroring, and centering.

As a consequence of this process, we effectively generate a much larger and variable number of examples from a single image in the dataset. Some patches will be closer to the native camera resolution and depict only part of the user face, while others will consist of most of the face, downscaled to the fixed input size. Different patches can emphasize different aspects of attack artifacts. As an effect, the trained network is expected to be more robust to variations in resolution. Moreover, the model must learn not to depend on certain combinations of facial features, which could naturally happen when training with aligned full faces.

**Training details.** We build mini-batches consisting of three columns of images from the same device: 64 genuine-access anchors, 64 genuine samples, and 64 attack samples; i.e., $T = 64$ in Eq 3. For each mini-batch, base images are sampled without replacement. Each triplet is passed through the network to calculate the aggregate maximum triplet loss component of the spoof loss, while the non-anchor samples are forwarded to the classification layer and used to compute the cross-entropy loss component. Gradients of the total *spoof loss* with respect to all parameters are then computed via backpropagation. Note that the triplet loss component does not influence the gradient at the classification layer.

The margin parameter $m$ in Eq 2 was set to 1.0 and we found little value in tweaking it, although, importantly, setting it to a large value can make optimization diverge in early iterations. The weighting parameter $\kappa$ in Eq 3 was set to 0.05. Starting at 0.5 and successively reducing it by 0.1 and then 0.05, we found this value to be the largest that consistently does not make training diverge in early stages.

We use the Adam optimizer with a learning rate of $10^{-5}$ and add an additional weight decay term with weight $10^{-4}$ to the final loss function.

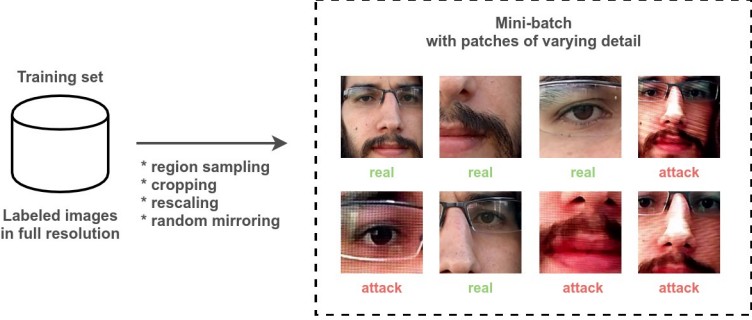

**Fig 5. Multi-resolution patches.** Mini-batch formation.

**Inference.** Crucially, given the network is fully-convolutional and is trained to be robust to variations in feature sizes, we can do fast and effective inference by using just a *single* image. We simply feed the network with a whole-face image.

## On-device user-specific adaptation

Thusfar we described how to train a CNN to solve the face PAD problem. In this section, we show how to further improve the effectiveness of these models in real-world situations, when they are deployed to mobile devices.

Classification models are trained with a finite training set, but are expected to work properly when presented with new data. Typically, if the operational data distribution is similar to the distribution of training data, models tend to behave well, but in practice there are no guarantees of generalization. Oftentimes, demands of the operational scenario are more specific. For example, in our case, the model will be deployed to a specific device, and will typically be presented with images from the same user.

Prior work in the literature [45, 46] proposed the learning of classifiers applied over user-specific features. This might not be appropriate for the mobile scenario, as the training would have to be performed on device. In face of this, we propose one simple strategy to adapt the decision boundary to the specific characteristics of the user and the sensor device.

During normal operation, and possibly over the course of many days, the user will have successfully authenticated multiple times, in different lighting situations. Perhaps even his appearance will have changed, due to changing glasses, haircut, facial hair, among other aspects. If we assume that after each successful authentication the final score (probability of attack) is stored in a user gallery, after some time we will have a representative distribution of the system's score for the genuine-access class. Alternatively, this gallery could be explicitly updated during enrollment sessions.

Our strategy takes advantage of a gallery $G$ of user-specific genuine-access scores stored in the device during normal operation. We assume that scores are in range $[0, 1]$ and that a higher score stands for higher likelihood of the input being an attack. We seek a minimal user-specific $\psi$, such that false rejection rate $FRR_\psi$ is bounded by a predefined value $\epsilon$.

**Algorithm 1** User-specific threshold estimation.

```
Input: score gallery G, tolerated FRR ε, threshold change Δ
Output: user-specific acceptance threshold ψ
  ψ ← 0
  while CummulativeDistribution(G, ψ) < 1 − ε do
    ψ ← ψ + Δ
  return ψ
```

Algorithm 1 describes the procedure. Function `CummulativeDistribution`($G$, $\psi$) returns the cummulative distribution of scores in the gallery, from 0 to $\psi$. For example, if $\epsilon = 0.05$, the search starts from 0 and stops at the first threshold $\psi$ for which no more than 5% of the genuine-access examples in the gallery would be rejected.

Despite its simplicity, the described procedure effectively tightens the decision boundary, so that the number of false rejections is controlled, while false acceptance errors are possibly reduced. In this way, a model is adjusted to make the best prediction it can for attacks, constrained to a certain user-inconvenience level $\epsilon$.

## Datasets

Building representative datasets is one of the hardest aspects of machine learning, being particularly difficult in this case, in which we deal with biometric data and we cannot account for all possible attack configurations. Furthermore, most of the existing public datasets do not fully

**Table 1. Summary of the datasets used in this work.**

| Property | RECOD-MPAD | OULU-NPU |
|---|---|---|
| Number of videos | 2,250 | 4,950 |
| Number of frames | 143,997 | variable |
| Number of users | 45 | 55 |
| Smartphone cameras | 2 (fixed focus) | 6 (fixed or auto-focus) |
| Pixel resolution | 1920 × 1080 | 1920 × 1080 |
| Number of sessions | **2 outdoors**, 3 indoors | 3 indoors |
| Session type | **dynamic** | static |
| Display attacks | **1 large**, 1 medium | 2 medium |
| Print attacks | 2 (**diff. lighting**) | 2 (**different printers**) |

satisfy the requirements, given our constraints and problem domain. Because of that, we focus on two datasets, RECOD-MPAD and OULU-NPU [34], which are described here in details. Table 1 summarizes the properties of the two datasets.

For the sake of completeness, we also experiment with two commonly used datasets in the literature, Idiap REPLAY-ATTACK [47] and CASIA Face AntiSpoofing [8], noting that they do not target the mobile environment. Idiap REPLAY-ATTACK contains 1200 videos captured by a MacBook webcam, under controlled and adverse conditions. Recaptures were recorded by a Canon Powershot camera, while the spoofing medium are an iPad 1, iPhone 3GS and paper. CASIA Face AntiSpoofing consists of 600 videos and considers more acquisitions devices than REPLAY-ATTACK, with different resolutions: a Sony NEX-5 camera and two USB cameras. Attacks are created with warped photos, cut photos and video.

## RECOD-MPAD

RECOD-MPAD was collected with the goal of building a truly representative dataset to our fast mobile-device unlock scenario, covering as many illumination variations as possible, as this is a lacking characteristic in public datasets. As such, it is the main benchmark for our method. The dataset is available at https://zenodo.org/record/3749309.

As the dataset consists of face images, all volunteers who agreed to participate were required to sign a term of consent allowing the use of their images for research purposes. The term was approved by the Research Ethics Committee of the University of Campinas under the number 53035216.6.0000.5404. Following common practices, data is anonymized in the sense that no additional information that could lead to the identification of the subjects is stored with the images. Specifically, subjects who appear in figures in this article have given additional written informed consent (as outlined in the PLOS consent form) to publish their image.

We used two acquisition devices: Moto G5 smartphone released in 2017 (device 1) and Moto X Style XT1572 smartphone from late 2015 (device 2). They are equipped with modern frontal cameras different from each another.

We designed five illumination scenarios for capturing the genuine-access videos:

- **Session 1**: Outdoors, direct sunlight on a sunny day.

- **Session 2**: Outdoors, in a shadow (diffuse lighting).

- **Session 3**: Indoors, artificial top light.

- **Session 4**: Indoors, natural lateral light (window or door).

- **Session 5**: Indoors, lights off (noisy).

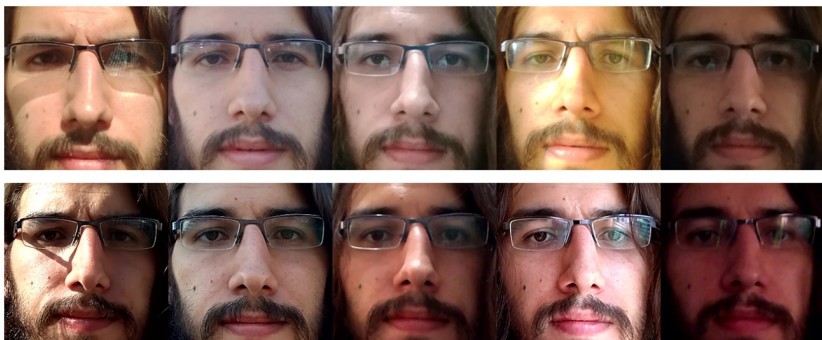

**Fig 6. RECOD-MPAD: Variations between genuine-access sessions and acquisition devices.** From left to right: sessions 1 to 5. Top row: acquisition device 1. Bottom row: acquisition device 2.

In contrast to the static sessions in previous datasets, each user was instructed to slowly rotate around his own axis during the roughly 10-second capture, further increasing variability from one frame to another.

The recaptures are divided into display and printed-photo attacks. For display attacks, we chose two monitors with different sizes as the attack medium—a large 42-inch monitor (D1) and a 17-inch monitor (D2); for printed-photo attacks, we extracted two frames from each of the original videos and printed them on A4-sized paper with a single printer. The first printout (P1) was recaptured in a scenario with diffuse lighting, while the second (P2) was recaptured in slightly dimmer and noisier conditions. Each recapturing session was done with the same corresponding acquisition device, following what would happen in a real-world setup. Fig 6 depicts some examples of genuine-access cropped frames and Fig 7 shows some recapture examples. To construct the official protocols, we extracted 64 equally-spaced frames from each video. These frames were then fed into the face landmark detector of the DLib toolkit (http://dlib.net/) and localized eye centers were saved. For frames in which the detection failed, we manually annotated eye centers. The final number of frames is 143,997, covering 45 users, 2 sensor devices, 5 sessions or illumination scenarios, and 4 attack types. Among the 45 users, 30 are men, 14 are wearing glasses, 13 have a beard, and age ranges from 18 to 50 years old. The frames are divided into 3 **user-disjoint** subsets:

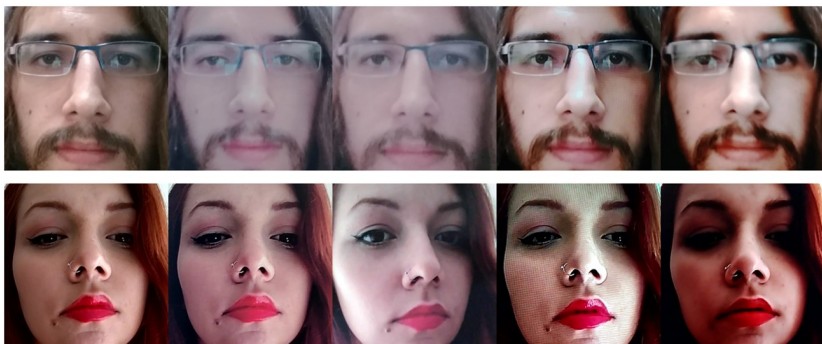

**Fig 7. RECOD-MPAD: Examples of display and printed-photo attacks.** From left to right: genuine, print 1, print 2, display 1, display 2. Top row: session 3. Bottom row: session 2.

- A training set containing 76,798 frames from 24 users;

- A validation or development set containing 19,200 frames from 6 users;

- A test set containing 47,999 frames from 15 users.

Using the locations of each eye, we align and crop each image to a square region by applying a similarity transform, so that the line connecting both eye centers is made horizontal and they occupy a standard position in the cropped face region. The transformation closely preserves the original resolution by mapping each eye so that the distance in pixels between them is roughly the same as in the unaligned image.

### OULU-NPU

The OULU-NPU dataset [34] was released for a face PAD competition targeting mobile devices [48]. It is is based on three static indoor sessions, and it includes genuine-access and attack videos taken with six different smartphone cameras.

Genuine-access videos were captured with the front camera of six smartphones. However, only two have normal fixed-focus frontal cameras. Three of them have auto-focusing capabilities, which dramatically influences close-distance recaptures. If a camera can properly focus on the attack surface, the result is a more detailed depiction of the original scene. But this can also emphasize details on the attack surface, resulting in aliasing or moiring artifacts.

In general, for each camera and user, three 5-second indoor videos were captured. Attacks were based on photos (printed on A3 paper using two different printers) and videos (recaptured on two different monitors).

Given that videos are mostly static, we extracted only 1 in 7 available frames, which results in 17 to 21 frames per video. We manually localized eyes in case annotations were missing. The remaining pre-processing is exactly as the one for RECOD-MPAD.

As, in this work, we generate scores for static frames, video scores for the OULU-NPU protocols are computed as the average score over the predicted frames. This makes comparison with other methods possible and fair.

## Experimental results

We report our results using a set of metrics commonly considered in biometric presentation attack evaluation, namely, Attack Presentation Classification Error Rate (*APCER*), Bona fide Presentation Classification Error Rate (*BPCER*), Average Classification Error Rate (*ACER*), Half-Total Error Rate (*HTER*), and Equal Error Rate (*EER*) [49].

$APCER_\theta$ and $BPCER_\theta$ are analogous to False Acceptance Rate (*FAR*) and False Rejection Rate (*FRR*), respectively, for a given acceptance threshold $\theta$. For the OULU-NPU evaluations, however, *APCER* also takes into account the attack potential in the worst-case scenario, i.e., *APCER* is the highest *FAR* computed for each presentation attack instrument (e.g., print and display) separately. *ACER* is the average of *APCER* and *BPCER*. Finally, *EER* is defined as the point where *FAR* is equal to *FRR*, and $HTER_\theta$ is the average of *FAR* and *FRR*, with acceptance threshold $\theta$.

In order to validate our proposal in RECOD-MPAD and OULU-NPU, we consider three baseline methods:

- **Whole-face CNN**: it consists of a convolutional neural network trained for the face PAD problem with whole-face images.

- **Pre-trained CNN**: it is a simplified version of the whole-face CNN, in which only the classification layer is learned, using the pre-trained frozen core as a feature extractor.

- **Color-LBP**: handcrafted method [50] that combines both texture and color characterizations, and was shown to outperform other popular texture-based methods.

For experiments using REPLAY-ATTACK and CASIA datasets, we consider as baseline a recent face presentation attack detection method based on unsupervised domain adaptation [51].

## Results on RECOD-MPAD

**Intra-scenario.** The proposed method and the baseline methods were trained with all available frames in the training set. Validation data were used in an initial experimentation phase to find suitable hyperparameters, which were then fixed. Validation data were also used to monitor training curves, and the final model is selected as the one that achieved the smallest $HTER_{0.5}$ in the validation set, after a certain number of epochs. Our method was trained for at most 300 epochs, while the other CNN baselines were trained for at most 100 epochs, as there was no observable benefit in continuing training beyond that point.

Table 2 presents the results, showing how well the tested methods can generalize to new users, but assuming similar acquisition and attack conditions. Our method outperformed the baselines by a large margin. Note that even a CNN whose core representations have been learned on another task can reach performance figures similar to those of the handcrafted baseline. This can also be validated by analyzing the trade-off between APCER and BPCER in the test set (Fig 8): our method outperforms the others mostly for all error values.

Fig 9 illustrates some error cases, with corresponding heatmaps showing which regions of the image are activated more strongly by the network. Although the examples show wrong prediction cases, we can observe, in the highlighted areas, that the network justifiably suggests clues of the opposite class. For instance, in Fig 9(c), the strong highlight caused by direct sunlight is marked as an attack clue, which makes sense. It suggests that errors are often interpretable and that the network is indeed learning useful representations that can discriminate attack and genuine frames.

**Cross-scenario.** In the cross-scenario experiment, our goal is to test how well models trained only in a limited number of scenarios can generalize to new conditions. This is one of the most overlooked aspects in PAD evaluation.

We start from the same user-disjoint subsets defined above. For the cross-session experiments, we create a total of 5 cross-session sub-protocols by using a leave-one-session-out strategy. For the cross-attack experiments, we filter the original subsets to create four sub-protocols: we select one type of display attack and one type of print attack to be part of the training and validation sets, while the test set is left only with the other two remaining attacks. For the

**Table 2. Results (%) on RECOD-MPAD.**

|  | Val. Set | Test Set | | |
|---|---|---|---|---|
| **Method** | **EER** | **$HTER_\theta$** | **$APCER_{0.5}$** | **$BPCER_{0.5}$** |
| Color-LBP [50] | 2.76 | 2.87 | 0.72 | 9.16 |
| Pre-trained CNN | 3.90 | 5.88 | 2.32 | 10.86 |
| Whole-face CNN | 0.94 | 1.08 | 0.35 | 1.29 |
| **Our method** | **0.37** | **0.54** | **0.12** | **1.15** |

Threshold $\theta$ corresponds to the EER in the validation set.

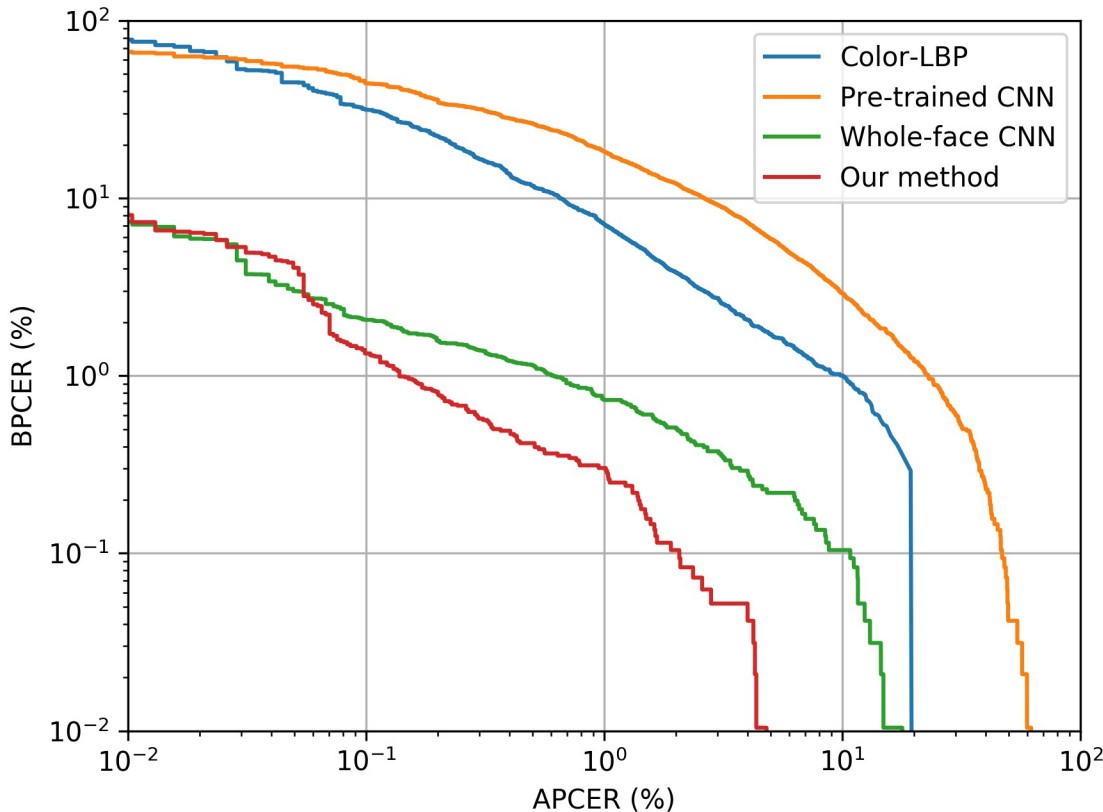

**Fig 8. Detection Error Trade-off (DET) for the methods evaluated on RECOD-MPAD.** Our method outperforms the others mostly for all error values.

cross-device experiments, we create two sub-protocols: each one has only frames from one device in the training and validation sets, and only frames from the other device in the test set.

Table 3 presents the results for the cross protocols. In general, the proposed method is comparable to or outperforms baselines when facing unseen conditions. In the cross-session protocol, our method is comparable to the Whole-face CNN baseline, with a much superior performance in the extreme low-light indoor scenario (5). In the cross-attack protocol, models trained with attacks from the larger display demonstrated acceptable generalization when predicting unseen attacks performed by the smaller monitor. Attacks performed with a smaller monitor and recaptured with a fixed-focus sensor tend to have limited resolution, due to soft focus at closer distances. For the cross-device protocol, the immediate observation is that models trained only with device 2 generalize relatively better than the other way around. Similarly to the situation with the cross-attack protocol, the most obvious difference is in the amount of detail (resolution) across different cameras and attack surfaces.

**On-device user-specific adaptation.**    Using RECOD-MPAD, we also tested the proposed on-device user-specific adaptation. Table 4 illustrates the benefits of using this technique. Two sessions are used to simulate a gallery, while errors are calculated considering the remaining three sessions. For this experiment, a different cut-off is learned for each of the 15 users in the test set, but we aggregate errors for convenience. Algorithm 1 is used with parameters $\epsilon = 0.05$ and $\delta = 0.05$.

By learning user-specific thresholds based on the recorded genuine-access scores from a user, we can obtain relative error reductions of up to about 30%, even when presented with

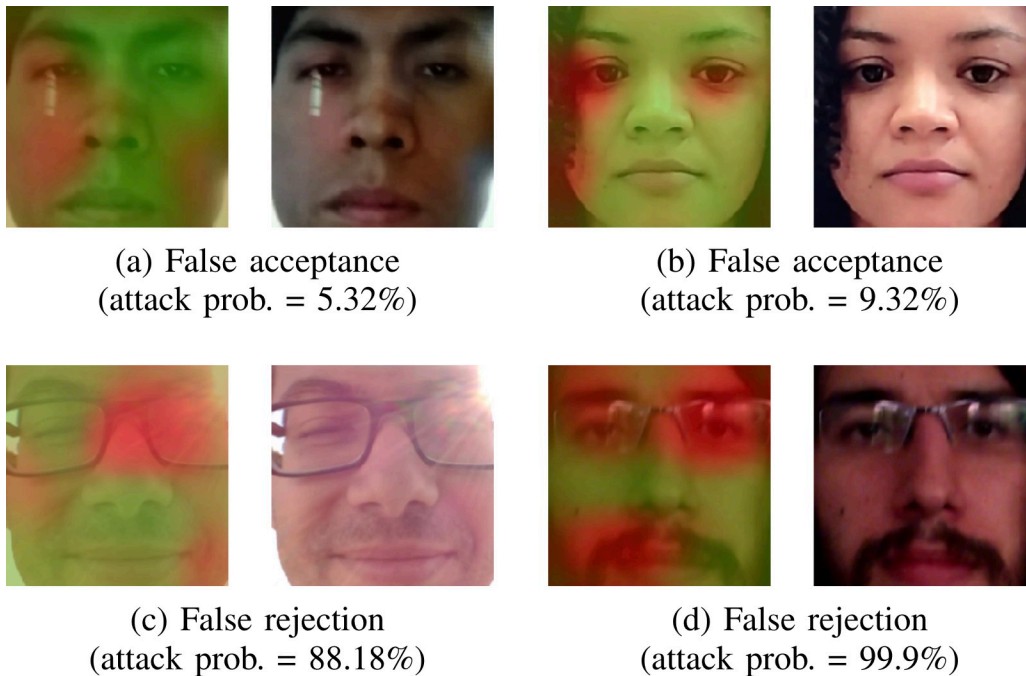

(a) False acceptance
(attack prob. = 5.32%)

(b) False acceptance
(attack prob. = 9.32%)

(c) False rejection
(attack prob. = 88.18%)

(d) False rejection
(attack prob. = 99.9%)

**Fig 9. Examples of error cases with corresponding heatmaps from the layer preceding global average pooling.** A brighter red hue stands for a locally higher likelihood of attack. These are difficult error cases, but some patterns indicate that the networks have learned useful features. In (a), the network correctly identifies the strong reflection as an attack clue, while the shadow areas in (b) contribute to the correct label. In (c), the strong highlight and loss of contrast caused by direct sunlight contributes to the wrong prediction, while in (d) the network gets confused by the reflection on the glasses and the overall blurriness, all of which are strong attack clues.

new attack types. The efficacy of the procedure could be higher in practice, in which the gallery could contain a more representative sample of illumination situations, instead of only two sessions needed for the experimental protocol.

## Results on OULU-NPU

The OULU-NPU dataset was used in an international competition with 13 participating teams [48], therefore it enabled us to compare our methods to the state of the art.

Table 5 summarizes the results for all protocols in the competition, each one designed to test a different cross scenario. Protocol IV is the hardest in the competition, as the test set includes unseen session, attacks, and user cameras. We compare our method first with the whole-face CNN baseline, and then with other competitors.

**Table 3. Results for the cross protocols of RECOD-MPAD.**

|  | Cross-session | | | | | Cross-attack | | | | Cross-device | |
| --- | --- | --- | --- | --- | --- | --- | --- | --- | --- | --- | --- |
| **Method** | **(1)** | **(2)** | **(3)** | **(4)** | **(5)** | **(D2 + P1)** | **(D1 + P2)** | **(D2 + P2)** | **(D1 + P1)** | **(1)** | **(2)** |
| Color-LBP [50] | 5.9 | 3.5 | 8.2 | 6.4 | 23.7 | 13.78 | 11.8 | 12.6 | 16.3 | 12.9 | 28.8 |
| Pre-trained CNN | 9.6 | 4.9 | 5.5 | 5.6 | 17.8 | 6.7 | 17.6 | 8.6 | 19.4 | 11.6 | 22.5 |
| Whole-face CNN | **0.8** | **0.5** | 6.1 | **1.7** | 16.5 | **4.6** | **11.4** | 4.7 | 13.0 | 8.7 | 16.6 |
| **Our method** | 3.0 | 1.9 | **5.1** | 3.0 | **11.8** | **4.6** | 12.3 | **2.9** | **9.9** | **6.0** | **8.8** |

Numbers are HTER$_\theta$ (%) in the test set, where $\theta$ is the EER threshold in the respective validation sets. In parentheses, the respective session, attacks or device in the test set.

**Table 4. User-specific adaptation.**

| Gallery | 1,2 | 1,3 | 1,4 | 1,5 | 2,3 | 2,4 | 2,5 | 3,4 | 3,5 | 4,5 |
|---|---|---|---|---|---|---|---|---|---|---|
| | Cross-attack (D2 + P1) | | | | | | | | | |
| $HTER_{0.5}$ | 5.3 | 4.2 | 5.0 | 4.5 | 4.6 | 5.4 | 5.0 | 4.3 | 3.8 | 4.6 |
| $HTER_{user}$ | 3.6 | 3.0 | 3.9 | 3.5 | 3.1 | 3.9 | 3.4 | 3.3 | 2.9 | 3.9 |
| **Error reduction (%)** | **33.3** | **27.7** | **25.9** | **22.5** | **33.2** | **28.6** | **30.9** | **22.2** | **24.3** | **14.2** |
| | Cross-attack (D1 + P2) | | | | | | | | | |
| $HTER_{0.5}$ | 11.7 | 9.4 | 11.4 | 14.4 | 9.4 | 11.5 | 14.5 | 9.2 | 12.2 | 14.2 |
| $HTER_{user}$ | 9.3 | 7.0 | 9.2 | 10.8 | 7.1 | 9.4 | 10.9 | 7.0 | 9.8 | 10.7 |
| **Error reduction (%)** | **19.8** | **24.7** | **18.8** | **24.8** | **25.0** | **18.2** | **24.6** | **23.4** | **19.3** | **24.2** |

In the first (cross-session) and second (cross-attack) protocols, our method performs similarly to the 3[rd] place, which utilizes Inception-V3 [52]. This network performs more operations than SqueezeNet, with a larger model size. In the third protocol (cross-device), our method does not perform well, as the dataset was constructed with cameras that are very different from one another. In the fourth and most challenging protocol (cross-*), the proposed method was on average better and, more importantly, only behind the first place.

It is important to note that the proposed method regularly performs above all other methods in the validation set, i.e., in an intra-dataset scenario. We also obtain an excellent result in the most challenging protocol, when all factors are unseen during training.

## Results on REPLAY-ATTACK and CASIA

To enable the comparison with other baselines in prior art, we experiment with two widely known datasets: Idiap REPLAY-ATTACK [47] and CASIA Face AntiSpoofing [8]. This also allows the evaluation of our method under a cross-dataset scenario, i.e., when training in one dataset and testing in another one, to evaluate generalization.

We compare our method to a recent state-of-the-art domain adaptation (DA) framework proposed for presentation attack detection [51]. In this work, the authors incorporated several features into the framework, however we present only the two features that generated better results: CoALBP HSV [3] and a deep-learning (DL) based feature [28]. For the cross-dataset scenario, the authors considered several domain adaptation techniques with outlier removal. Here, we present results with the Kernel Subspace Alignment (KSA) method, which provided better results in general.

Table 6 summarizes the results. Our method provides better results in all cases, expect for the cross-dataset experiment when training with REPLAY-ATTACK and testing on CASIA.

**Table 5. Results (%) for the OULU-NPU dataset.**

| Method | Protocol I cross-session | | Protocol II cross-attack | | Protocol III cross-device | | Protocol IV cross-* | |
|---|---|---|---|---|---|---|---|---|
| | Val. EER | Test ACER | Val. EER | Test ACER | Val. EER | Test ACER | Val. EER | Test ACER |
| Whole-face CNN | 2.6 | **8.3** | 3.7 | 11.3 | 4.5 ± 0.9 | 19.7 ± 5.8 | 3.9 ± 0.3 | 19.2 ± 10.5 |
| **Our method** | **0.8** | 8.8 | **0.7** | **7.2** | **0.6 ± 0.3** | **13.6 ± 7.0** | **0.5 ± 0.3** | **14.2 ± 6.1** |
| Competition baseline | 4.4 | 12.9 | 4.1 | 14.6 | 3.9 ± 0.7 | 11.4 ± 4.6 | 4.7 ± 0.6 | 26.3 ± 16.9 |
| GRADIANT (1st) | 0.7 | **6.5** | 0.9 | **2.5** | 0.9 ± 0.4 | **3.8 ± 2.4** | 1.1 ± 0.3 | **10.0 ± 5.0** |
| Massy-HNU (2nd) | **0.6** | 6.9 | 1.3 | 6.1 | 1.4 ± 0.5 | 6.5 ± 6.4 | **1.0 ± 0.4** | 22.1 ± 17.6 |
| CPqD (3rd) | 2.2 | 8.3 | 4.4 | 6.7 | 0.9 ± 0.4 | 7.4 ± 3.3 | 2.2 ± 1.7 | 22.1 ± 20.8 |

We report each value with its corresponding standard devition σ over sub-protocols, if applicable.

**Table 6. Results for CASIA (EER), REPLAY-ATTACK ($HTER_{0.5}$), and cross-dataset scenarios ($HTER_{0.5}$).**

| Method | CASIA | REPLAY-ATTACK | CASIA $\Rightarrow$ REPLAY-ATTACK | REPLAY-ATTACK $\Rightarrow$ CASIA |
|---|---|---|---|---|
| DA [51] w/ CoALBP HSV [3] | 5.5 | 3.7 | 35.1 | 39.8 |
| DA [51] w/ CoALBP HSV [3] | 7.6 | 2.1 | 39.3 | **12.3** |
| Our method | **4.8** | **1.5** | **30.8** | 53.9 |

In this specific setup, the training dataset contains only one capturing device (a webcam), which hinders the application of our method. We also highlight that the domain adaptation framework and the datasets were not proposed for the mobile scenario and they do not take into account its pecularities. Nevertheless, as expected, the error rates are higher in the cross-dataset scenario, for all methods, as capturing conditions strongly differ from the training to the test set.

## Mobile implementation

The proposed method were implemented for Android, using TensorFlow [53]. The implementation was tested in the previously considered devices: Moto G5 smartphone released in 2017 (device 1) and Moto X Style XT1572 smartphone from late 2015 (device 2). Each one has a 8-core 2.0 GHz CPU.

In device 1, the method runs in 197.75 ± 34.57ms, while in device 2 the running time is 233.85 ± 28.55ms, calculated over 20 independent runs. The peak memory usage is 50MB. This does not include face detection and alignment, but that is negligible in comparison to the forward pass.

The proposed method can run in modern smartphone devices in under one second, and the small amount of required memory ensures that the authentication step does not interfere with other applications.

## Conclusions

In this work, we proposed a new method to train a CNN to model the face PAD problem in a completely data-driven way. This allowed us to gain insight into the problem definition itself, instead of being tied to specific handcrafted features. We focused on the constraints of the mobile-device scenario, with its data acquisition peculiarities and hardware limitations.

The novel formulation seeks to improve upon the traditional interpretation of the problem: binary classification of aligned faces. The use of patches of varying resolution during training, besides increasing the number of available examples, also forces the model to be robust to changes in resolution and to avoid overfitting to specific facial features. We also proposed a multi-objective loss function, specially designed to the problem, to estimulate genuine-access examples from the same device to be more compactly located in the learned feature space, while also reducing inter-device confusion, an issue that has not been addressed in the literature.

To train the models and evaluate the solutions, we introduced a new dataset—RECOD-M-PAD—with unique characteristics, including low-light and outdoor lighting scenarios, with much higher intra- and inter-session variability than existing datasets. It also defines challenging factor-disjoint protocols, a problem often overlooked in previous protocols.

Our approach proved to be superior or comparable to handcrafted methods and other CNN baselines. This suggests that, contrary to popular belief, complex deep learning models can generalize better than handcrafted alternatives, even when trained with arguably limited

amounts of data. The proposed method was also evaluated on a public dataset that was part of a competition with 13 competing solutions, to which we compare favorably.

Crucially, the trained architecture has very small memory requirements, and can make predictions within a fraction of a second in modern smartphones. This validates the potential of data-driven approaches in solving the presentation attack detection problem in mobile environments.

Apart from being applicable to low-end devices, which do not contain advanced sensors, our method could be combined or extended with different types of inputs, including depth information, thus taking advantage of new technologies. Furthermore, the proposed loss function is generic enough to be applied to other PAD problems, forcing the genuine-access examples from the same capturing device to be more compactly located in the learned feature space, while reducing inter-device confusion.

In future investigations, we intend to experiment with other CNN architectures and analyze the impact of multi-resolution patch inputs and the spoof loss function when training them. We also want to study the use of multi-resolution patches during inference, instead of whole-face images. By augmenting the inference for an input with its multi-resolution patches, we can compute a fusion score which is hypothetically more robust. In this case, we need to further investigate the trade-offs between mobile processing time and the improvements induced by the fusion process.

## Author Contributions

**Conceptualization:** Waldir R. Almeida, Fernanda A. Andaló, Rafael Padilha, Ricardo da S. Torres, Jacques Wainer, Anderson Rocha.

**Data curation:** Waldir R. Almeida, Fernanda A. Andaló, Rafael Padilha, Gabriel Bertocco, William Dias.

**Formal analysis:** Waldir R. Almeida, Fernanda A. Andaló, Ricardo da S. Torres, Jacques Wainer, Anderson Rocha.

**Funding acquisition:** Anderson Rocha.

**Investigation:** Fernanda A. Andaló, Rafael Padilha, Gabriel Bertocco, William Dias, Ricardo da S. Torres, Jacques Wainer, Anderson Rocha.

**Methodology:** Waldir R. Almeida, Fernanda A. Andaló, Rafael Padilha, Ricardo da S. Torres, Jacques Wainer, Anderson Rocha.

**Project administration:** Anderson Rocha.

**Resources:** Anderson Rocha.

**Supervision:** Fernanda A. Andaló, Ricardo da S. Torres, Jacques Wainer, Anderson Rocha.

**Validation:** Waldir R. Almeida, Fernanda A. Andaló, Ricardo da S. Torres, Jacques Wainer, Anderson Rocha.

**Writing – original draft:** Waldir R. Almeida, Fernanda A. Andaló.

**Writing – review & editing:** Waldir R. Almeida, Fernanda A. Andaló, Rafael Padilha, Gabriel Bertocco, William Dias, Ricardo da S. Torres, Jacques Wainer, Anderson Rocha.

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
