## [Decision Letter · Decision Letter 0]

16 Mar 2020

PONE-D-20-01807

Detecting face presentation attacks in mobile devices with a patch-based CNN and a sensor-aware loss function

PLOS ONE

Dear Dr. Andalo,

Thank you for submitting your manuscript to PLOS ONE. After careful consideration, we feel that it has merit but does not fully meet PLOS ONE’s publication criteria as it currently stands. Therefore, we invite you to submit a revised version of the manuscript that addresses the points raised during the review process.

We would appreciate receiving your revised manuscript by Apr 30 2020 11:59PM. To enhance the reproducibility of your results, we recommend that if applicable you deposit your laboratory protocols in protocols.io, where a protocol can be assigned its own identifier (DOI) such that it can be cited independently in the future. For instructions see: http://journals.plos.org/plosone/s/submission-guidelines#loc-laboratory-protocols

We look forward to receiving your revised manuscript.

Kind regards,

He Debiao

Academic Editor

PLOS ONE

Journal Requirements:

2. Were any images collected specifically for this study? Please provide additional details regarding participant consent to collect personal data and images. In the Methods section, please ensure that you have specified how consent was obtained and how the study met relevant personal data and privacy laws. If data were collected anonymously, please include this information. Please also indicate whether there was any ethical oversight over the study, and please include this information in both the Ethics statement and manuscript. Please also include information on how participants were recruited, and whether they consented to have their images shared publicly.

4.We note that Figure 2, 5, 6, 7 and 9 includes an image of a participant in the study. 

Please respond by return e-mail with an amended manuscript. We can upload this to your submission on your behalf.

If you are unable to obtain consent from the subject of the photograph, please either instruct us to remove the figure or supply a replacement figure by return e-mail for which you hold the relevant copyright permissions and subject consents. In some cases, you may need to specify in the text that the image used in the figure is not the original image used in the study, but a similar image used for illustrative purposes only. We can make any changes on your behalf.

5. We note that Figure 2, 5, 6, 7 and 9 in your submission contain copyrighted images. All PLOS content is published under the Creative Commons Attribution License (CC BY 4.0), which means that the manuscript, images, and Supporting Information files will be freely available online, and any third party is permitted to access, download, copy, distribute, and use these materials in any way, even commercially, with proper attribution. For more information, see our copyright guidelines: http://journals.plos.org/plosone/s/licenses-and-copyright.

1.    You may seek permission from the original copyright holder of  to publish the content specifically under the CC BY 4.0 license.

Reviewers' comments:

Reviewer's Responses to Questions

**Comments to the Author**

1. Is the manuscript technically sound, and do the data support the conclusions?

Reviewer #1: Yes

Reviewer #2: Yes

2. Has the statistical analysis been performed appropriately and rigorously? 

Reviewer #1: Yes

Reviewer #2: N/A

3. Have the authors made all data underlying the findings in their manuscript fully available?

Reviewer #1: Yes

Reviewer #2: No

4. Is the manuscript presented in an intelligible fashion and written in standard English?

Reviewer #1: Yes

Reviewer #2: Yes

5. Review Comments to the Author

Reviewer #1: Strengths:

1)Authors have efficiently used SqueezeNet as the base of the training model which has around 1.2 million parameters that takes only 5MB of memory which is very suitable for a mobile device to operate.

2)The model structure is fully convolutional which made the input size and alignment much flexible and more accurate than AlexNet. All feature maps are flexible in size, but they kept the input shape as 3 x 227 x 227. And that gives one important flexibility. Scaling-down the pixels of the images does not lose much localisation information, rather it lets us free from taking long time to train the model.

3)Authors used variable resolution face-patches of the images. They claimed that it reduces overfitting to user-specific characteristics. And thus the learning becomes more robust as the images does not remain tied to any single scale anymore.

4)Through the experiment they found out that the camera specifications are very important here and while building the model this should be considered to avoid ending up with a biased model. And to overcome this problem, they came up with triplet-loss which also can be called intra-device triplet loss and spoof-loss which also can be called multi-objective loss instead of traditional cross-entropy loss.

5)Authors expressed their experiment result using 5 common considered biometric presentation attack evaluation namely Attack Presentation Classification Error Rate (APCER), Bona fide Presentation Classification Error Rate (BPCER), Average Classification Error Rate (ACER), Half-total Error Rate (HTER) and Equal Error Rate (EER). This made their experiment much stronger.

Weaknesses:

1)In the datasets authors have used, the devices were used are a little old version. The latest phone they have used Moto G5 was released in 2017. But in those phones there was no face detection technology. Rather face detection technologies are provided in the phones those were released very recently and obviously cameras on those phone are different. So those new phones like iphone 10, 11, galaxy s10 etc should have been given priority to acquire data. Also iPad should have been included in the device list.

2)Authors didn’t mentioned about gender or racial diversity in the dataset. There should have been a proper diversity ratio in the dataset.

3)Authors should have shown not only images but how video streams performs with this model.

4)Authors should have shown what properties of other CNN structures are failing in mobile device environment while SqueezeNet architecture is winning the situation with proper use-cases and example situations.

5)The main idea of this experiment was to differentiate between genuine access images and fake attack images. But ideally this experiment can be done not only on face images rather on images of other materials. They could show some results of experiments on the images other than face.

Reviewer #2: This paper presents a CNN-based algorithm to detect face presentation attacks, specifically for mobile devices. The idea is to use patches of varying resolution. This not only increases the number of available examples but also solves the overfitting issues. The authors also introduce a multi-objective loss function to reduce inter-device confusion. A comprehensive evaluation has been conducted on RECOD-MPAD, a new dataset tailored to the mobile-device setup. The results show that the proposed method achieves better results compared with state-of-the-art solutions.

Strength:

1: The paper is clearly written and well structured.

2: A new dataset is developed. It will be great if the authors can make it publicly available.

3: Evaluation is comprehensive and well justified.

Weakness:

Novelty is somehow limited. My understanding is that the authors merely apply the existing SqueezeNet architecture to the mobile scenario.

The authors claim that the dataset will be made available along with the article publication. It cannot be verified at the review stage.

Minor:

Page 2 line 42 - 49: missing section numbers.

Page 6 line 220 - 223: missing section numbers.

6. PLOS authors have the option to publish the peer review history of their article (what does this mean?). If published, this will include your full peer review and any attached files.

Reviewer #1: No

Reviewer #2: No

---

## [Author Response · Author response to Decision Letter 0]

3 Jun 2020

We provide, as a separate document, a detailed list of all reviewers’ recommendations and our answers.

---

## [Decision Letter · Decision Letter 1]

26 Jun 2020

PONE-D-20-01807R1

Detecting face presentation attacks in mobile devices with a patch-based CNN and a sensor-aware loss function

PLOS ONE

Dear Dr. Andalo,

Thank you for submitting your manuscript to PLOS ONE. After careful consideration, we feel that it has merit but does not fully meet PLOS ONE’s publication criteria as it currently stands. Therefore, we invite you to submit a revised version of the manuscript that addresses the points raised during the review process.

We look forward to receiving your revised manuscript.

Kind regards,

He Debiao

Academic Editor

PLOS ONE

Reviewers' comments:

Reviewer's Responses to Questions

**Comments to the Author**

1. If the authors have adequately addressed your comments raised in a previous round of review and you feel that this manuscript is now acceptable for publication, you may indicate that here to bypass the “Comments to the Author” section, enter your conflict of interest statement in the “Confidential to Editor” section, and submit your "Accept" recommendation.

Reviewer #1: All comments have been addressed

Reviewer #2: (No Response)

2. Is the manuscript technically sound, and do the data support the conclusions?

Reviewer #1: Yes

Reviewer #2: Yes

3. Has the statistical analysis been performed appropriately and rigorously? 

Reviewer #1: Yes

Reviewer #2: N/A

4. Have the authors made all data underlying the findings in their manuscript fully available?

Reviewer #1: Yes

Reviewer #2: No

5. Is the manuscript presented in an intelligible fashion and written in standard English?

Reviewer #1: Yes

Reviewer #2: Yes

6. Review Comments to the Author

Reviewer #1: The authors have addressed all comments satisfactorily. I recommend acceptance. This could be an important paper in the community of computer vision and security.

Reviewer #2: Thanks the authors for the contribution clarification.

Another concern I have after reading the revision is that the technique proposed in the paper is limited to images captured by traditional cameras. However, with the development of new technologies, such as face ID, which creates a depth map of faces and also captures an infrared image, the proposed approach will soon be obsolete. There is no doubt that new hardware can achieve better accuracy and privacy. The authors can argue that the proposed approach can be used in low-end smartphone. Other than that, could the authors provide any other insights?

7. PLOS authors have the option to publish the peer review history of their article (what does this mean?). If published, this will include your full peer review and any attached files.

Reviewer #1: No

Reviewer #2: No

---

## [Author Response · Author response to Decision Letter 1]

14 Jul 2020

The response to the reviewers is detailed in a separate document.

---

## [Decision Letter · Decision Letter 2]

10 Aug 2020

Detecting face presentation attacks in mobile devices with a patch-based CNN and a sensor-aware loss function

PONE-D-20-01807R2

Dear Dr. Andalo,

We’re pleased to inform you that your manuscript has been judged scientifically suitable for publication and will be formally accepted for publication once it meets all outstanding technical requirements.

Kind regards,

He Debiao

Academic Editor

PLOS ONE

Additional Editor Comments (optional):

Reviewers' comments:

Reviewer's Responses to Questions

**Comments to the Author**

1. If the authors have adequately addressed your comments raised in a previous round of review and you feel that this manuscript is now acceptable for publication, you may indicate that here to bypass the “Comments to the Author” section, enter your conflict of interest statement in the “Confidential to Editor” section, and submit your "Accept" recommendation.

Reviewer #1: All comments have been addressed

Reviewer #2: All comments have been addressed

2. Is the manuscript technically sound, and do the data support the conclusions?

Reviewer #1: Yes

Reviewer #2: Yes

3. Has the statistical analysis been performed appropriately and rigorously? 

Reviewer #1: Yes

Reviewer #2: Yes

4. Have the authors made all data underlying the findings in their manuscript fully available?

Reviewer #1: Yes

Reviewer #2: Yes

5. Is the manuscript presented in an intelligible fashion and written in standard English?

Reviewer #1: Yes

Reviewer #2: Yes

6. Review Comments to the Author

Reviewer #1: The authors address all concerns. The paper is recommended for acceptance. The response from the authors is thorough.

Reviewer #2: The authors have addressed all my concerns, including the discussion of advanced sensors and the release of the dataset. I recommend acceptance.

7. PLOS authors have the option to publish the peer review history of their article (what does this mean?). If published, this will include your full peer review and any attached files.

Reviewer #1: No

Reviewer #2: No

---

## [Editor Report · Acceptance letter]

14 Aug 2020

PONE-D-20-01807R2 

Detecting face presentation attacks in mobile devices with a patch-based CNN and a sensor-aware loss function 

Dear Dr. Andalo:

I'm pleased to inform you that your manuscript has been deemed suitable for publication in PLOS ONE. Congratulations! Your manuscript is now with our production department. 

Kind regards, 

on behalf of

Dr. He Debiao 

Academic Editor

PLOS ONE